# Enhancing Trust-Region Bayesian Optimization via Derivatives of Gaussian Processes

## Abstract

Bayesian Optimization (BO) has been widely applied to optimize expensive black-box functions while retaining sample efficiency. However, scaling BO to high-dimensional spaces remains challenging. Existing literature proposes performing standard BO in several local trust regions (TuRBO) for heterogeneous modeling of the objective function and avoiding over-exploration. Despite its advantages, using local Gaussian Processes (GPs) reduces sampling efficiency compared to a global GP. To enhance sampling efficiency while preserving heterogeneous modeling, we propose to construct several local quadratic models using gradients and Hessians from a global GP, and select new sample points by solving the bound-constrained quadratic program. We provide a convergence analysis and demonstrate through experimental results that our method enhances the efficacy of TuRBO and outperforms a wide range of high-dimensional BO techniques on synthetic functions and real-world applications.

## 1 Introduction

Bayesian Optimization (BO) has been one of the popular methods for the global optimization of expensive black-box functions due to its high sampling efficiency. Applications include hyperparameter tuning for deep learning (Hvarfner et al., 2022), discovering new molecules for chemical engineering (Gómez-Bombarelli et al., 2018), searching an optimal policy for reinforcement learning (Müller et al., 2021), and so on. BO is a sequential model-based approach consisting of two main components: a surrogate model and an acquisition function. The surrogate model, typically implemented as a Gaussian Process regression, is used to improve the sampling efficiency of BO by modeling the objective function. The acquisition function is used to determine the next sample point.

While BO performs well in optimizing low-dimensional functions, it struggles with high-dimensional problems for several reasons. First, the surrogate model loses accuracy in the high-dimensional space when estimating the objective function. This is because it is impossible to fill the high-dimensional space with finite sample points, even with a large sample size (Györfi et al., 2002). Second, the computational complexity of optimizing the acquisition function grows exponentially with dimensions (Kandasamy et al., 2015).

Various methods have been proposed to address the curses of dimensionality in BO. The vast majority of the prior work assumes special structures in the objective function, such as additive structure (Kandasamy et al., 2015; Han et al., 2021) or intrinsic dimension (Wang et al., 2016; Letham et al., 2020). However, these assumptions are often too restrictive for widespread application. Other works directly improve the high-dimensional BO without additional assumptions, including TuRBO (Eriksson et al., 2019), GIBO (Müller et al., 2021), and MPD (Nguyen et al., 2022).

This paper focuses on trust-region Bayesian Optimization (TuRBO). TuRBO is attractive because it uses local GPs for heterogeneous modeling of the objective function and performs BO locally in several trust regions to avoid over-exploration. However, using local GPs reduces sampling efficiency compared to a global GP. To overcome this limitation, we propose a new trust-region BO method (TuRBO-D) that incorporates the derivatives of GPs. It constructs several local quadratic models using gradients and Hessians from a global GP, enabling heterogeneous modeling of the objective function while maintaining the same sample efficiency of a global GP. To optimize globally, it maintains multiple trust regions simultaneously. Our method consists of three main stages: building

several local quadratic models using derivatives from a global GP, selecting new sample points by solving the bound-constrained quadratic program in each trust region, and updating the trust region radii based on new evaluations. In addition, we provide theoretical proof that our method converges to stationary points with high probability. In summary, our main contributions are:

- Proposing a new trust-region BO method that incorporates GP derivatives to enhance sampling efficiency while retaining heterogeneous modeling.
- Providing a convergence analysis guaranteeing the convergence of our proposed method.
- Empirically validating our method on synthetic and real-world applications, demonstrating improved efficacy over TuRBO and outperforming various high-dimensional BO methods.

## 2 RELATED WORK

In the realm of high-dimensional BO, there are generally three kinds of methods. The first kind of method assumes the existence of a lower-dimensional structure within objective functions, typically employing a three-stage process: producing a low-dimensional embedding, performing standard BO in this low-dimensional space, and projecting found optimal points back to the original space. In REMBO (Wang et al., 2016), the low-dimensional embedding is achieved by using a random projection matrix. But REMBO often produces points that fall outside the box bounds of the original space, necessitating their projection onto the facet of the box and resulting in a harmful distortion. Subsequently, several techniques are proposed to fix this problem (Letham et al., 2020; Binois et al., 2020). In addition, the random low-dimensional embedding can be also achieved by randomized hashing functions (Nayebi et al., 2019; Papenmeier et al., 2022). The key advantage of the hashing functions lies in their ability to effortlessly map candidate points back to the original space, thus circumventing the need for clipping to box-bound facets. Some works achieve linear embeddings based on learning. For example, SIR-BO employs Sliced Inverse Regression to derive the linear embeddings, while SI-BO (Djolonga et al., 2013) learns the linear embeddings via low-rank matrix recovery. Garnett et al. (2014) learn the linear embeddings by maximizing the marginal likelihood of GPs. Besides, nonlinear embedding techniques have also been explored, particularly those based on Variational Autoencoders (Gómez-Bombarelli et al., 2018; Lu et al., 2018). However, these approaches typically require a substantially larger sample size. In addition to embedding techniques, some research has focused on variable selection methods (Kirschner et al., 2019; Li et al., 2017; Shen & Kingsford, 2023; Song et al., 2022).

The second kind of method assumes the existence of an additive structure for the objective function. The additive objective function can be modeled by additive GPs (Kandasamy et al., 2015), allowing for more efficient maximization of the acquisition function. However, the true additive structure still remains challenging to learn. Several works propose to learn the underlying additive structure from training data. For example, Wang et al. (2017) proposed a method that employs the Dirichlet process to assign input variables into distinct groups. Rolland et al. (2018) employ a dependency graph to model the interactions between input variables, allowing for the assignment of input variables into overlapping groups. Han et al. (2021) proposed a refinement that restricts the dependency graph to a tree structure, reducing the computational complexity of maximizing acquisition functions. In contrast to data-driven decomposition methods, RDUCB (Ziomek & Bou-Ammar, 2023) learns a random tree-based decomposition to mitigate the potential mismatch between the objective function and additive GPs.

The third kind of method focuses on direct enhancements to the BO process in high-dimensional spaces, without relying on any other assumption. For example, TuRBO (Eriksson et al., 2019), GIBO (Müller et al., 2021) and MPD (Nguyen et al., 2022) adopt local strategies for BO to avoid over-exploration in high-dimensional spaces. Another set of approaches focuses on partitioning the search space and identifying a promising region to perform BO more efficiently (Wang et al., 2014; Kawaguchi et al., 2015; Wang et al., 2020). Researchers have also proposed better initialization methods for optimizing high-dimensional acquisition functions efficiently (Rana et al., 2017; Zhao et al., 2024).

GIBO and MPD are similar to ours, which also utilize gradients of GPs. In contrast to their work, our work incorporates both gradient and Hessian information from GPs and provides a convergence analysis.

## 3 BACKGROUND

### 3.1 BAYESIAN OPTIMIZATION

Bayesian optimization considers an optimization problem $\min_{\mathbf{x}\in\mathcal{X}} f(\mathbf{x})$ where $f$ is a black-box and derivative-free function over a hyper-rectangular feasible set $\mathcal{X}$. As a sequential model-based approach, BO comprises two main components: a surrogate model and an acquisition function. The surrogate model approximates the objective function, while the acquisition function, based on this model, determines the next sampling point. Gaussian Process regression is typically employed as the surrogate model (Rasmussen & Williams, 2006), $f \sim \mathcal{GP}(m(\cdot), k(\cdot, \cdot))$ with a mean function $m(\cdot)$ and a kernel $k(\cdot, \cdot)$. More specifically, GP assumes that evaluations of any finite number sampling point $\mathbf{x}_{1:n}$ follow a joint Gaussian distribution, $\mathbf{f} \sim \mathcal{N}(\mathbf{m}(\mathbf{x}_{1:n}), \mathbf{K}(\mathbf{x}_{1:n}, \mathbf{x}_{1:n}))$. Given training data $\mathcal{D}_n = \{\mathbf{x}_{1:n}, \mathbf{y}_{1:n}\}$ and a new point $\mathbf{x}_*$, the joint distribution is given by

$$\begin{bmatrix} \mathbf{y}_{1:n} \\ f(\mathbf{x}_*) \end{bmatrix} \sim \mathcal{N}\left( \begin{bmatrix} \mathbf{m}(\mathbf{x}_{1:n}) \\ m(\mathbf{x}_*) \end{bmatrix}, \begin{bmatrix} \mathbf{K}(\mathbf{x}_{1:n}, \mathbf{x}_{1:n}) + \sigma_n^2 \mathbf{I} & \mathbf{k}(\mathbf{x}_{1:n}, \mathbf{x}_*) \\ \mathbf{k}(\mathbf{x}_*, \mathbf{x}_{1:n}) & k(\mathbf{x}_*, \mathbf{x}_*), \end{bmatrix} \right)$$

where $\sigma_n^2$ is the variance of Gaussian noise added to the observations. It follows from the Sherman-Morrison-Woodbury formula that the posterior normal distribution for $f(\mathbf{x}_*)$ is given by $f(\mathbf{x}_*)|\mathcal{D}_n, \mathbf{x}_* \sim \mathcal{N}(\mu_n(\mathbf{x}_*), \sigma_n^2(\mathbf{x}_*))$ where

$$\mu_n(\mathbf{x}_*) = m(\mathbf{x}_*) + \mathbf{k}(\mathbf{x}_*, \mathbf{x}_{1:n})(\mathbf{K}(\mathbf{x}_{1:n}, \mathbf{x}_{1:n}) + \sigma_n^2 \mathbf{I})^{-1}(\mathbf{y}_{1:n} - \mathbf{m}(\mathbf{x}_{1:n}))$$

$$\sigma_n^2(\mathbf{x}_*) = k(\mathbf{x}_*, \mathbf{x}_*) - \mathbf{k}(\mathbf{x}_*, \mathbf{x}_{1:n})(\mathbf{K}(\mathbf{x}_{1:n}, \mathbf{x}_{1:n}) + \sigma_n^2 \mathbf{I})^{-1}\mathbf{k}(\mathbf{x}_{1:n}, \mathbf{x}_*)$$

Based on this posterior, an acquisition function $\alpha(\cdot)$ is constructed to quantify the utility of sampling points. Common choices include Expected Improvement (Jones et al., 1998) and Entropy Search (Hennig & Schuler, 2012). The next sample point is determined by maximizing the acquisition function, $\mathbf{x}_{n+1} = \arg\max_{\mathbf{x}\in\mathcal{X}} \alpha(\mathbf{x})$. After evaluating the objective function at $\mathbf{x}_{n+1}$, the process advances to the next iteration.

### 3.2 UNIFORM ERROR BOUNDS OF THE GP

Under the mild assumption of Lipschitz continuity for both the objective function and the kernel function, a directly computable probabilistic uniform error bound can be established.

**Assumption 1.** *The unknown objective function $f$ is a sample from a Gaussian process $\mathcal{GP}(0, k(\mathbf{x}, \mathbf{x}'))$ and observations are perturbed by Gaussian noise, $y = f(\mathbf{x}) + \epsilon$, where $\epsilon \sim \mathcal{N}(0, \sigma^2)$. The unknown function $f$ is continuous with the Lipschitz constant $L_f$ and the kernel $k$ is Lipschitz continuous with the Lipschitz constant defined as*

$$L_k := \max_{\mathbf{x}, \mathbf{x}' \in \mathcal{X}} \left\| \left( \frac{\partial k(\mathbf{x}, \mathbf{x}')}{\partial x_1}, \dots, \frac{\partial k(\mathbf{x}, \mathbf{x}')}{\partial x_D} \right)^\top \right\|_2 .$$

**Theorem 1** (Theorem 3.1 in (Lederer et al., 2019)). *Given an unknown function $f$ satisfying Assumption 1, the posterior mean function $\mu_t$ from the GP fitted on the training data $\mathcal{D}_t$ is continuous with the Lipschitz constant $L_{\mu_t}$, and the standard deviation $\sigma_t$ admits a modulus of continuity $\omega_{\sigma_t}$ on $\mathcal{X}$, where*

$$L_{\mu_t} \leq L_k \sqrt{t} \|(\mathbf{K} + \sigma_t^2 \mathbf{I})^{-1} \mathbf{y}\|_2$$

$$\omega_{\sigma_t}(\tau) \leq \sqrt{2\tau L_k \left( 1 + t \|(\mathbf{K} + \sigma_t^2 \mathbf{I})^{-1}\|_2 \max_{\mathbf{x}, \mathbf{x}' \in \mathcal{X}} k(\mathbf{x}, \mathbf{x}') \right)}.$$

*Moreover, given $\delta \in (0, 1)$, $\tau > 0$, one has that*

$$\mathbb{P}\left( |f(\mathbf{x}) - \mu_t(\mathbf{x})| \leq \sqrt{\beta(\tau)} \sigma_t(\mathbf{x}) + \gamma(\tau), \ \forall \mathbf{x} \in \mathcal{X} \right) \geq 1 - \delta, \tag{1}$$

*where*

$$\beta(\tau) = 2 \log \left( \frac{M(\tau, \mathcal{X})}{\delta} \right), \quad \gamma(\tau) = (L_{\mu_t} + L_f)\tau + \sqrt{\beta(\tau)} \omega_{\sigma_t}(\tau),$$

*and $M(\tau, \mathcal{X})$ is the covering number that is the minimum number of spherical balls with radius $\tau$ required to completely cover $\mathcal{X}$.*

## 4 METHOD

In this section, we propose a novel trust-region BO method for optimizing high-dimensional black-box functions. To address the reduced sampling efficiency of local GPs in TuRBO, we construct several local quadratic models using gradients and Hessians from a global GP. This approach allows for heterogeneous modeling of the objective function while maintaining the same sample efficiency of a global GP. To achieve global optimization, we select new sample points by solving the bound-constrained quadratic programs in multiple regions.

**Local modeling.** At iteration $k$, with $\mathbf{x}_k$ as the best solution found so far, the local quadratic model is defined as,

$$m_k(\mathbf{x}_k + \mathbf{s}) = f(\mathbf{x}_k) + \mathbf{g}_k^\top \mathbf{s} + \frac{1}{2}\mathbf{s}^\top \mathbf{B}_k \mathbf{s}, \tag{2}$$

where $\mathbf{g}_k$ and $\mathbf{B}_k$ approximate the gradient and Hessian of the objective function, respectively. Since the derivatives of the objective function $f$ are unknown, we set

$$\mathbf{g}_k = \nabla \mu_k(\mathbf{x}_k), \quad \mathbf{B}_k = \nabla^2 \mu_k(\mathbf{x}_k) + \lambda \nabla^2 \sigma_k(\mathbf{x}_k),$$

where $\lambda$ is a hyperparameter, $\mu_k(\cdot)$ and $\sigma_k(\cdot)$ are the posterior mean and standard deviation of the GP model.

**Trust regions.** To ensure the quadratic model $m_k$ accurately approximates $f$, $\mathbf{x}_k + \mathbf{s}$ needs to be restricted to a trust region $\mathcal{B}_k$ defined as

$$\mathcal{B}_k := \{\mathbf{x} \in \mathbb{R}^D \mid \|\mathbf{x} - \mathbf{x}_k\| \le \Delta_k\},$$

where $\Delta_k$ is the trust-region radius, adjusted iteratively. It should be decreased when the optimizer appears stuck and increased when the optimizer finds better solutions. When the radius falls below a predetermined minimum threshold $\Delta_{\min}$, it signals that the current region has been thoroughly explored. At this point, the algorithm restarts in another region to promote global exploration. In this paper, we adopt the same radius update strategy as TuRBO, which has proven effective in balancing local exploitation and global exploration.

**Trust regions in the $\infty$-norm.** In BO, the search space is typically a rectangular box. Without loss of generality, we assume that the box is $[0, 1]^D$. Given this constraint, the trust region is defined as

$$\mathcal{B}_k := \{\mathbf{x} \in \mathbb{R}^D \mid \|\mathbf{x} - \mathbf{x}_k\| \le \Delta_k, \, \mathbf{0} \le \mathbf{x} \le \mathbf{1}\}.$$

When the trust region is in the Euclidean norm, $\mathcal{B}_k$ consists of the intersection of a sphere and a rectangular (Jorge & Stephen, 2006), leading to more complex quadratic models. To simplify this, we adopt the $\infty$-norm for the trust region, which transforms $\mathcal{B}_k$ into a simple rectangular,

$$\mathcal{B}_k := \{\mathbf{x} \in \mathbb{R}^D \mid -\Delta_k \mathbf{1} \le \mathbf{x} - \mathbf{x}_k \le \Delta_k \mathbf{1}, \, \mathbf{0} \le \mathbf{x} \le \mathbf{1}\}.$$

Then candidate is selected by solving the bound-constrained quadratic program,

$$\underset{\mathbf{s}}{\text{minimize}} \quad m_k(\mathbf{x}_k + \mathbf{s}), \quad \text{subject to } \mathbf{x}_k + \mathbf{s} \in \mathcal{B}_k. \tag{3}$$

The above problem can be solved by gradient projection methods. However, the Hessian of the GP is often nearly singular, which can lead to issues when using conjugate gradient iterations. Such methods may require numerous iterations and yield only small reductions in each step. Instead, we employ a gradient projection method using quasi-Newton iterations, specifically L-BFGS-B (Byrd et al., 1995). This approach approximates the singular Hessian with a positive definite matrix, improving the efficiency and robustness of the optimization process.

**Derivatives vanish in the high-dimensional space.** In general, our approach is effective for medium-dimensional problems (typically $D < 100$). However, as the dimensionality increases beyond this range, the derivatives of GPs tend to vanish, posing a significant challenge to our method. To mitigate this issue and ensure the derivatives remain informative, we choose $d$ variables out of $D$

variables randomly as the working set $\mathcal{W}_k$ at each iteration. Then, a global GP is constructed on the working set and the bound-constrained quadratic program is denoted as

$$
\begin{aligned}
\underset{\mathbf{s}}{\text{minimize}} \quad & m_k(\mathbf{x}_k + \mathbf{s}) = f(\mathbf{x}_k) + \mathbf{g}_k^\top \mathbf{s} + \frac{1}{2}\mathbf{s}^\top \mathbf{B}_k \mathbf{s}, \\
\text{subject to} \quad & \mathbf{s}_i = 0, \ \forall i \notin \mathcal{W}_k \\
& \mathbf{x}_k + \mathbf{s} \in \mathcal{B}_k.
\end{aligned}
\tag{4}
$$

So far, we have detailed a single local BO strategy using a trust region. To achieve global optimization in this framework, we maintain $m$ trust regions simultaneously, selecting a candidate within each trust region to form a batch of $m$ candidates. We denote our method as TuRBO-D, as presented in Algorithm 1.

---

**Algorithm 1:** TuRBO-D

---

**Input:** $n$, $T$, $M$
**Output:** The sample points and their evaluations $\mathcal{D}_T$
1 $\mathcal{D}_0 = \{\mathbf{x_{1:n}}, \mathbf{y_{1:n}}\} \leftarrow$ Randomly sample $n$ points from the feasible set $\mathcal{X}$ and then evaluate these points;
2 **Initializations.** Choose an initial radius for each trust region, $\{\Delta_0^{(\ell)}\}_{\ell=1}^M$, and determine an initial point for each trust region, $\{\mathbf{x}_0^{(\ell)}\}_{\ell=1}^M \subset \mathcal{D}_0$;
3 **for** $k \leftarrow 1$ **to** $T$ **do**
4 $\quad$ Build a global GP based on the training data $\mathcal{D}_k$;
5 $\quad$ **for** $\ell \leftarrow 1$ **to** $M$ **do**
6 $\quad\quad$ Build a local quadratic model $m_k^{(\ell)}(\mathbf{x}_k^{(\ell)} + \mathbf{s})$ in the $\ell$-th trust region;
7 $\quad\quad$ Select a candidate by minimizing the model within the $\ell$-th trust region according to Eq.3;
8 $\quad\quad$ Evaluate the candidate, $y_{k+1}^{(\ell)} \leftarrow f(\mathbf{x}_{k+1}^{(\ell)})$;
9 $\quad\quad$ Update the trust-region radius $\Delta_k^{(\ell)}$ based on new evaluations;
10 $\quad$ Update the training data, $\mathcal{D}_{k+1} \leftarrow \mathcal{D}_k \cup \{\mathbf{x}_{k+1}^{(\ell)}, y_{k+1}^{(\ell)}\}_{\ell=1}^M$;
11 **return** $\mathcal{D}_T$

---

## 5 A CONVERGENCE ANALYSIS

Our method shares several key features with trust-region derivative-free optimization methods, including the use of quadratic models to approximate the objective function and adaptive trust region updates. However, a crucial distinction lies in the nature of the error between the quadratic model and the objective function. This error is probabilistic in our approach, while it is typically deterministic in derivative-free optimization methods using interpolation techniques. This probabilistic aspect necessitates a verification of the coherence between the derivatives of GPs and those of the objective function. This fundamental difference precludes the direct application of standard convergence theory for derivative-free methods to our method. Consequently, we must reconsider the convergence analysis in detail.

To maintain analytical simplicity, we adopt the same assumptions as (Conn et al., 1997) and follow their trust region update strategy, as outlined in Algorithm 2.

**Assumption 2.** *The objective function $f : \mathbb{R}^D \to \mathbb{R}$ is twice continuously differentiable whose gradient $\nabla f(\mathbf{x})$ and Hessian $\nabla^2 f(\mathbf{x})$ is uniformly bounded in the norm. In other words, there are constants $\kappa_{fg} > 0$ and $\kappa_{fh} > 0$ such that*

$$
\|\nabla f(\mathbf{x})\|_2 \leq \kappa_{fg}, \quad \|\nabla^2 f(\mathbf{x})\|_2 < \kappa_{fh}
$$

*for all $\mathbf{x} \in \mathbb{R}^D$.*

**Assumption 3.** *The objective function is bounded below on $\mathbb{R}^D$.*

**Assumption 4.** *The approximate Hessians $\mathbf{B}_k$ are uniformly bounded in the norm. In other words, there is a constant $\kappa_{mh} > 0$ such that $\|\mathbf{B}_k\|_2 \leq \kappa_{mh}, \ \forall \mathbf{x} \in \mathcal{B}_k$.*

**Algorithm 2:** The trust-region update strategy in derivative-free optimization

**Input:** $\mathbf{s}_k$, $\Delta_k$, $0 < \eta_0 \leq \eta_1 < 1$, $0 < \beta_1 < 1 < \beta_2$, $\mu \geq 1$

**Output:** $\Delta_{k+1}$

1 Compute the ratio

$$\rho_k := \frac{f(\mathbf{x}_k) - f(\mathbf{x}_k + \mathbf{s}_k)}{m_k(\mathbf{x}_k) - m_k(\mathbf{x}_k + \mathbf{s}_k)}.$$

**if** $\rho_k \geq \eta_1$ **then**

2 $\quad$ $$\Delta_{k+1} \leftarrow \min\{\beta_2 \Delta_k, \mu\|\mathbf{g}_k\|_2\}.$$

3 **else if** $\rho_k < \eta_0$ **then**

4 $\quad$ $$\Delta_{k+1} \leftarrow \beta_1 \Delta_k.$$

5 **else**

6 $\quad$ $\Delta_{k+1} \leftarrow \Delta_k$;

7 **return** $\Delta_{k+1}$

---

**Lemma 1** (Lemma 6 in (Conn et al., 1997)). *At every iteration $k$, one has that*

$$m_k(\mathbf{x}_k) - m_k(\mathbf{x}_k + \mathbf{s}_k) \geq \kappa_{mdc}\|\mathbf{g}_k\| \min\left(\Delta_k, \frac{\|\mathbf{g}_k\|}{\kappa_h}\right),$$

*for some constant $\kappa_{mdc} \in (0,1)$ independent of $k$, where $\kappa_h = \max\{\kappa_{fg}, \kappa_{fh}, \kappa_{mh}\}$.*

**Theorem 2.** *Assume that Assumption 1, 2, and 4 hold. Then given $\delta \in (0,1)$ there is $\kappa_{em}$ such that*

$$\mathbb{P}\left(|f(\mathbf{x}) - m_k(\mathbf{x})| \leq \kappa_{em} \max\{\Delta_k, \Delta_k^2\}, \ \forall \mathbf{x} \in \mathcal{B}_k \ \forall k\right) \geq 1 - \delta.$$

*Proof.* It follows from Taylor's theorem that

$$f(\mathbf{x}_k + \mathbf{s}) = f(\mathbf{x}_k) + \nabla f(\mathbf{x}_k)^\top \mathbf{s} + \int_0^1 [\nabla f(\mathbf{x}_k + t\mathbf{s}) - \nabla f(\mathbf{x}_k)]^\top \mathbf{s} \, dt,$$

for some $t \in (0,1)$. Then

$$|m_k(\mathbf{x}_k + \mathbf{s}) - f(\mathbf{x}_k + \mathbf{s})|$$

$$= \left| [\nabla\mu_k(\mathbf{x}_k) - \nabla f(\mathbf{x}_k)]^\top \mathbf{s} + \frac{1}{2}\mathbf{s}^\top \nabla^2\mu_k(\mathbf{x}_k)\mathbf{s} - \int_0^1 [\nabla f(\mathbf{x}_k + t\mathbf{s}) - \nabla f(\mathbf{x}_k)]^\top \mathbf{s} \, dt \right|$$

$$\leq \|\nabla\mu_k(\mathbf{x}_k) - \nabla f(\mathbf{x}_k)\|_2 \|\mathbf{s}\|_2 + (\kappa_{mh}/2)\|\mathbf{s}\|_2^2 + \kappa_{fh}\|\mathbf{s}\|_2^2 \tag{5}$$

It follows from Eq. 1 that

$$\mathbb{P}\left(\|\nabla\mu_k(\mathbf{x}_k) - \nabla f(\mathbf{x}_k)\|_2 \leq \sqrt{\beta(\tau)}\|\nabla\sigma_k(\mathbf{x}_k)\|_2, \forall k\right) \geq 1 - \delta.$$

In fact, assume without loss of generality that $f(\mathbf{x}_k) - \mu_t(\mathbf{x}_k) \leq \sqrt{\beta(\tau)}\sigma_t(\mathbf{x}_k) + \gamma(\tau)$, then following the continuity of $f(\mathbf{x})$, $\mu_t(\mathbf{x})$ and $\sigma_t(\mathbf{x})$, there is $\varepsilon \in (0,1)$ such that $\forall i \in \{1, \ldots, D\}$

$$f(\mathbf{x}_k + \varepsilon\mathbf{e}_i) - \mu_t(\mathbf{x}_k + \varepsilon\mathbf{e}_i) \leq \sqrt{\beta(\tau)}\sigma_t(\mathbf{x}_k + \varepsilon\mathbf{e}_i) + \gamma(\tau).$$

Hence, combing the above two inequalities, one has that

$$\frac{f(\mathbf{x}_k + \varepsilon\mathbf{e}_i) - f(\mathbf{x}_k)}{\varepsilon} - \frac{\mu_t(\mathbf{x}_k + \varepsilon\mathbf{e}_i) - \mu_t(\mathbf{x}_k)}{\varepsilon} \leq \sqrt{\beta(\tau)}\frac{\sigma_t(\mathbf{x}_k + \varepsilon\mathbf{e}_i) - \sigma_t(\mathbf{x}_k)}{\varepsilon}.$$

Letting $\varepsilon \to 0$, one has that

$$\frac{\partial f(\mathbf{x}_k)}{\partial x_i} - \frac{\partial\mu_t(\mathbf{x}_k)}{\partial x_i} \leq \sqrt{\beta(\tau)}\frac{\partial\sigma_t(\mathbf{x}_k)}{\partial x_i}.$$

Similarly, if $\mu_t(\mathbf{x}_k) - f(\mathbf{x}_k) \leq \sqrt{\beta(\tau)}\sigma_t(\mathbf{x}_k) + \gamma(\tau)$, then

$$\frac{\partial \mu_t(\mathbf{x}_k)}{\partial x_i} - \frac{\partial f(\mathbf{x}_k)}{\partial x_i} \leq \sqrt{\beta(\tau)}\frac{\partial \sigma_t(\mathbf{x}_k)}{\partial x_i}, \ \forall i \in \{1 \ldots D\}.$$

Since then, it has been proved the event $|f(\mathbf{x}_k) - \mu_t(\mathbf{x}_k)| \leq \sqrt{\beta(\tau)}\sigma_t(\mathbf{x}_k) + \gamma(\tau)$ implies that $\|\nabla\mu_k(\mathbf{x}_k) - \nabla f(\mathbf{x}_k)\|_2 \leq \sqrt{\beta(\tau)}\|\nabla\sigma_k(\mathbf{x}_k)\|_2$.

Since $\sigma_k$ admits a modulus of continuity according to Theorem 1, there is $\kappa_{eg}$ such that $\|\nabla\sigma_k(\mathbf{x}_k)\|_2 \leq \kappa_{eg}\Delta_k$. Then

$$\mathbb{P}\left(\|\nabla\mu_k(\mathbf{x}_k) - \nabla f(\mathbf{x}_k)\|_2 \leq \kappa_{eg}\sqrt{\beta(\tau)}\Delta_k, \forall k\right) \geq 1 - \delta. \tag{6}$$

Combining Eq. 5 and 6, one has that

$$\mathbb{P}\left[|m_k(\mathbf{x}_k + \mathbf{s}) - f(\mathbf{x}_k + \mathbf{s})| \leq (\kappa_{eg}\sqrt{\beta(\tau)} + \kappa_{mh}/2 + \kappa_{fh})\max\{\Delta_k, \Delta_k^2\}, \ \forall k\right] \geq 1 - \delta$$

Hence, $\kappa_{em} = \kappa_{eg}\sqrt{\beta(\tau)} + \kappa_{mh}/2 + \kappa_{fh}$. □

**Lemma 2.** *Assume that Assumption 1-4 hold. In addition, assume that there is a constant $\kappa_g > 0$ such that $\|g_k\| \geq \kappa_g$ for all $k$. Then given $\delta \in (0, 1)$ there is a constant $\kappa_d$ such that*

$$\mathbb{P}\left(\Delta_k > \kappa_d, \ \forall k\right) \geq 1 - \delta.$$

*Proof.* It follows from Lemma 7 in (Conn et al., 1997) that if $|f(\mathbf{x}) - m_k(\mathbf{x})| \leq \kappa_{em}\max\{\Delta_k, \Delta_k^2\}$, then $\forall k, \ \Delta_k > \kappa_d$, where

$$\kappa_d = \beta_1 \min\left(1, \frac{\kappa_{mdc}\kappa_g(1 - \eta_1)}{\max(\kappa_h, \kappa_{em})}\right).$$

And since it follows from Theorem 2 that

$$\mathbb{P}\left(|f(\mathbf{x}) - m_k(\mathbf{x})| \leq \kappa_{em}\max\{\Delta_k, \Delta_k^2\}, \ \forall \mathbf{x} \in \mathcal{B}_k \ \forall k\right) \geq 1 - \delta.$$

and hence, we obtain

$$\mathbb{P}\left(\Delta_k > \kappa_d, \ \forall k\right) \geq 1 - \delta.$$

□

This property ensures that the radius cannot become too small with a high probability as long as the gradient of the GP does not vanish.

**Theorem 3.** *Assume that Assumption 1-4 hold. Then it holds that*

$$\liminf_{k \to \infty} \|\mathbf{g}_k\|_2 = 0$$

*Proof.* We proceed by contradiction. Suppose there is $\kappa_g > 0$ such that $\|\mathbf{g}_k\| \geq \kappa_g$ for all $k$. It follows from Theorem 9 in (Conn et al., 1997) that if $\Delta_k > \kappa_d$ for all $k$, then

$$f(\mathbf{x}_0) - f(\mathbf{x}_{k+1}) \geq \frac{1}{2}\sigma_k\kappa_g\eta_0 \min\left(\frac{\kappa_g}{\kappa_h}, \kappa_d\right)$$

where $\sigma_k$ is the number of successful iterations up to iteration $k$. In our case, it follows from Lemma 2 that

$$\mathbb{P}\left(\Delta_k > \kappa_d, \forall k\right) \geq 1 - \delta.$$

This implies that

$$\mathbb{P}\left(f(\mathbf{x}_0) - f(\mathbf{x}_{k+1}) \geq \frac{1}{2}\sigma_k\kappa_g\eta_0 \min\left(\frac{\kappa_g}{\kappa_h}, \kappa_d\right)\right) \geq 1 - \delta.$$

And since $\lim_{k \to \infty} \sigma_k = +\infty$, one has that $\forall M \in \mathbb{R} \ \exists k$,

$$\mathbb{P}\left(f(\mathbf{x}_0) - f(\mathbf{x}_{k+1}) > M\right) \geq 1 - \delta,$$

which contradicts the fact that $f$ is bounded. □

**Lemma 3.** *Assume that Assumption 1-4 hold. If there is a subsequence $\{k_i\}$ such that* $\lim_{i \to \infty} \|\mathbf{g}_{k_i}\| = 0$*, then given $\delta \in (0, 1)$ it holds that $\forall \epsilon \in (0, 1) \; \exists N$,*

$$\mathbb{P}\left(\|\nabla f(\mathbf{x}_{k_i})\|_2 < \epsilon, \; \forall i > N\right) \geq 1 - \delta.$$

*Proof.* It follows from Eq. 6 that

$$\mathbb{P}\left(\|\nabla f(\mathbf{x}_{k_i}) - \mathbf{g}_{k_i}\|_2 \leq \kappa_{eg}\sqrt{\beta(\tau)}\Delta_{k_i}, \; \forall i\right) \geq 1 - \delta.$$

And since $\Delta_{k_i} \leq \mu \|\mathbf{g}_{k_i}\|_2$ (according to Algo. 2), one has that

$$\mathbb{P}\left(\|\nabla f(\mathbf{x}_{k_i}) - \mathbf{g}_{k_i}\|_2 \leq \kappa_{eg}\sqrt{\beta(\tau)}\mu\|\mathbf{g}_{k_i}\|_2, \; \forall i\right) \geq 1 - \delta.$$

And since $\|\nabla f(\mathbf{x}_{k_i})\|_2 \leq \|\mathbf{g}_{k_i}\|_2 + \|\nabla f(\mathbf{x}_{k_i}) - \mathbf{g}_{k_i}\|_2$, one has that

$$\mathbb{P}\left(\|\nabla f(\mathbf{x}_{k_i})\|_2 \leq (1 + \kappa_{eg}\sqrt{\beta(\tau)}\mu)\|\mathbf{g}_{k_i}\|_2, \; \forall i\right) \geq 1 - \delta.$$

Combining the limit $\lim_{i \to \infty} \|\mathbf{g}_{k_i}\|_2 = 0$ and the above equation, one has that $\forall \epsilon \in (0, 1) \; \exists N$,

$$\mathbb{P}\left(\|\nabla f(\mathbf{x}_{k_i})\|_2 < \epsilon, \forall i > N\right) \geq 1 - \delta.$$

$\square$

**Theorem 4.** *Assume that Assumption 1-4 hold. Then given $\delta \in (0, 1)$, there is a sequence of iterations $\{\mathbf{x}_k\}$ such that $\forall \epsilon \in (0, 1) \; \exists N$,*

$$\mathbb{P}\left(\inf_{k > N} \|\nabla f(\mathbf{x}_k)\| = 0\right) \geq 1 - \delta.$$

*Proof.* The result immediately follows from Theorem 3 and Lemma 3. $\square$

The theorem ensures that our approach will converge to stationary points with a high probability.

## 6 EXPERIMENTAL RESULTS

In this section, we evaluate our method (TuRBO-D) on a wide range of benchmarks: 50-dimensional synthetic functions, 100-dimensional synthetic functions, a 300-dimensional Lasso tuning problem, a 180-dimensional Lasso tuning problem, and a 124-dimensional vehicle design problem.

We compare our method (TuRBO-D) to a broad selection of existing methods: linear embedding methods (ALEBO (Letham et al., 2020), SIR-BO), nonlinear embedding methods (KSIR-BO (Zhang et al., 2019)), BO using additive models (Add-GP-UCB (Kandasamy et al., 2015)), local-search methods (TuRBO, GIBO), and quasirandom search (Sobol). For BO using embedding, we take $d = 10$ for these experiments. For Add-GP-UCB, we take $d = 4$ for each group. TuRBO-D and TuRBO maintain 5 trust regions simultaneously. In 100-dimensional synthetic functions, Lasso and MOPTA08, we choose 50 variables randomly as the working set at each iteration for TuRBO-D to ensure derivatives of GPs remain informative. We test all methods using 50 initial points and batch size of $q = 5$.

### 6.1 SYNTHETIC EXPERIMENTS

First, we consider the 50-dimensional Ackley function in the domain $[-5, 10]^{50}$, and the 50-dimensional Griewank function in the domain $[-300, 600]^{50}$. Both functions feature numerous local minima and a global minimum, making them suitable for testing global optimization methods. Fig. 1 shows that TuRBO-D enhances the efficacy of TuRBO and gets the best performance of all methods on the mid-dimensional synthetic functions. The initialization strategy of ALEBO favors sampling points away from the boundary, resulting in high-quality initial samples. However, the optimizer of ALEBO tends to stagnate when objective functions lack lower-dimensional structure. SIR-BO and KSIR-BO demonstrate poor performance in this problem, yielding results comparable

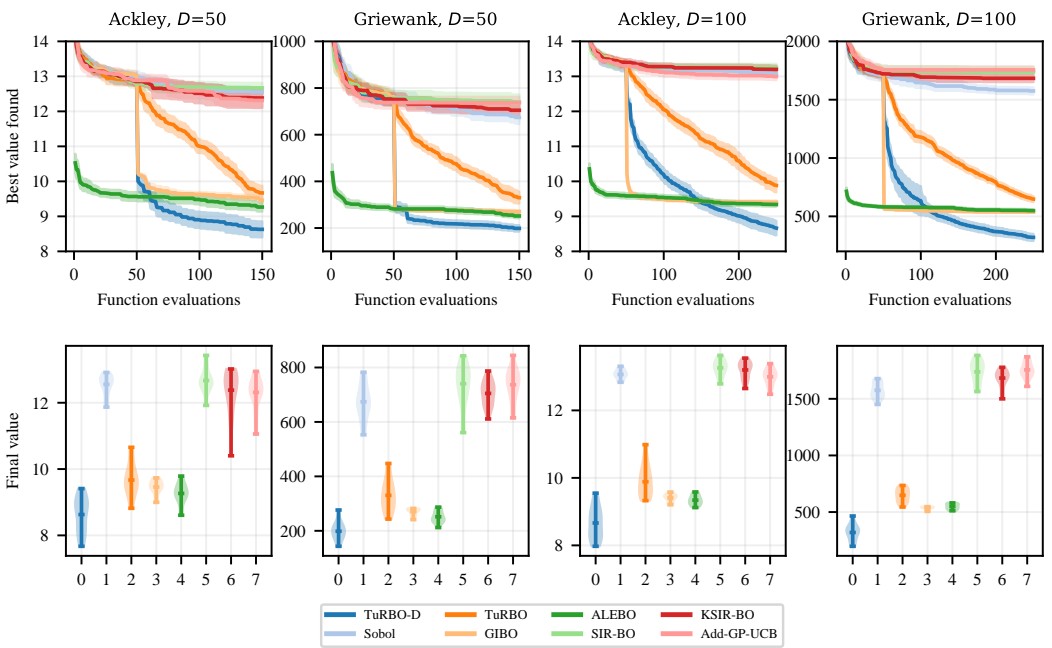

Figure 1: We compare TuRBO-D to baseline methods on 50-dimensional functions and 100-dimensional functions, showing (Top row) optimal values by each iteration averaged over 20 repeated runs, and (Bottom row) the distribution over the final optimal values over 20 repeated runs.

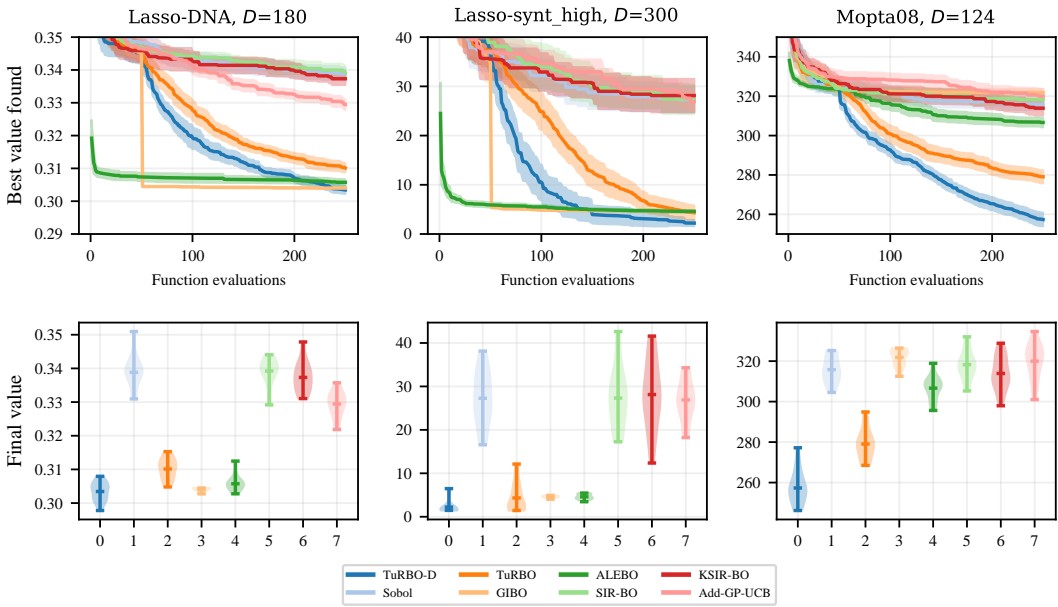

Figure 2: We compare TuRBO-D to baseline methods on the Lasso-DNA tuning ($D = 180$), Lasso-synt_high tuning ($D = 300$) and MOPTA vehicle design ($D = 124$), showing (Top row) optimal values by each iteration averaged over 20 repeated runs, and (Bottom row) the distribution over the final optimal values over 20 repeated runs.

to random search. Add-GP-UCB also underperforms on this problem because objective functions lack additive structure.

Second, we consider the 100-dimensional Ackley function in the domain $[-5, 10]^{100}$, and the 50-dimensional Griewank function in the domain $[-300, 600]^{100}$. Fig. 1 shows that TuRBO-D again enhances the efficacy of TuRBO and gets the best performance among all methods on the high-dimensional synthetic functions. GIBO always samples the midpoint of the domain after initialization. It suffers from the vanishing gradients of GPs in the high-dimensional spaces, causing it to become stuck at the midpoint. ALEBO once again encounters stagnation after initialization due to the absence of lower-dimensional structure in these functions. SIR-BO, KSIR-BO and Add-GP-UCB underperform on these high-dimensional functions without lower-dimensional structure or additive structure.

## 6.2 REAL-WORLD PROBLEMS

**Weighted Lasso Tuning.**   We consider the problem of tuning the Lasso (Least Absolute Shrinkage and Selection Operator) regression models. LassoBench (Sehic et al., 2022) provides a set of benchmark problems for tuning penalty terms for Lasso models. In Lasso, each regression coefficient corresponds to a penalty term, so the number of hyperparameters equals the number of features in the dataset. We focus on two Lasso tuning problems: a 180-dimensional DNA dataset with 43 effective dimensions, and a 300-dimensional synthetic dataset with 15 effective dimensions.

Fig. 2 shows that TuRBO-D enhances the efficacy of TuRBO and achieves the best performance among all methods on the Lasso-synt_high problem. For the Lasso-DNA problem, TuRBO-D eventually attains optimal values comparable to GIBO while outperforming other methods. GIBO, after initially sampling the midpoint, stagnates due to vanishing gradients of GPs in high-dimensional spaces. Its performance is primarily attributed to this initial midpoint sampling. ALEBO also becomes stuck after initialization, despite the existence of lower-dimensional structure in these problems. SIR-BO and KSIR-BO perform poorly, yielding results comparable to random search. Interestingly, Add-GP-UCB shows better performance than SIR-BO and KSIR-BO, despite LassoBench lacking the additive structure that Add-GP-UCB typically exploits.

**Vehicle Design.**   We consider the vehicle design problem with a soft penalty as defined in (Eriksson & Jankowiak, 2021). The objective is to minimize the mass of a vehicle characterized by 124 design variables describing materials, gauges, and vehicle shape. This results in a 124-dimensional optimization problem.

Fig. 2 shows that TuRBO-D enhances the efficacy of TuRBO and achieves the best performance among all methods on the MOPTA08 problem. In this problem, the midpoint is not close to the optimal point, resulting in initial strategies of GIBO and ALEBO performing comparably to random search. ALEBO outperforms the other embedding approaches on the MOPTA08, while GIBO stagnates and performs worse than random search. SIR-BO, KSIR-BO and Add-GP-UCB underperform on the MOPTA08 due to its lack of lower-dimensional structure or additive structure.

## 7 CONCLUSION

In this paper, we introduce TuRBO-D, a novel trust-region BO method that incorporates the derivatives of GPs for enhancing the sampling efficiency of TuRBO. This novel scheme is realized by (1) constructing several local quadratic models using gradients and Hessians from a global GP, enabling heterogeneous modeling of the objective function while maintaining the same sample efficiency of a global GP, and (2) selecting new sample points by solving the bound-constrained quadratic program in multiple trust regions. Comprehensive experimental evaluations demonstrate that TuRBO-D significantly enhances the efficacy of TuRBO and outperforms a wide range of high-dimensional BO methods on a set of synthetic functions and three real-world applications. Furthermore, we provide a convergence analysis for our method.

While we mitigate the problem of vanishing derivatives using working sets, we will focus on developing better schemes to address this challenge in the future.

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
