# OpenReview forum: "Enhancing Trust-Region Bayesian Optimization via Derivatives of Gaussian Processes"
_ICLR.cc/2025/Conference — ICLR 2025 Conference Withdrawn Submission_

### Official Review · Reviewer_gnTJ · 2024-10-29

**Soundness:** 1
**Presentation:** 2
**Contribution:** 2
**Rating:** 3
**Confidence:** 3

**Summary:**

This paper considers the high-dimensional Bayesian optimization problem.
The proposed algorithm, TuRBO-D, extends the existing TuRBO algorithm by incorporating a quadratic model into the local optimization routine of TuRBO.
By combining the existing convergence analysis of trust-region derivative-free optimization methods
with the probabilistic error bounds of GP sample paths, the authors show high-probability convergence
to the stationary point of the proposed local optimization routine.
The empirical performance of the proposed algorithm is confirmed with 2 synthetic and 3 real-world-based problems.

**Strengths:**

- Originality: Although the basic algorithm construction relies on the existing TuRBO algorithm, incorporating global GP modeling information into the local quadratic optimization routine is novel to me.
- Quality: The theoretical analysis and proofs support the effectiveness of the proposed method; however, I believe that more discussion about the assumptions of this paper is essential.

**Weaknesses:**

- The validity of Assumptions 2, 3, and 4 is not discussed. I believe that these assumptions are not valid under the assumption $f \sim \mathcal{GP}(0, k)$ (suggested in Assumption 1), even if we adopt commonly-used SE or Matérn family kernels. Some assumptions seem to be valid for these commonly-used kernels by relying on more careful probabilistic arguments; however, others are unknown to me. I summarize my current understanding below:
    - Assumption 3: When the kernel is SE or Matérn, the boundedness of the sample path is valid with high probability if the input domain is compact (e.g., Ghosal & Roy (2006), Kandasamy et al. (2019)); however, the same result does not hold for $\mathbb{R}^D$ (as the authors assumed).
    - Assumption 1: When the kernel is four times continuously differentiable and stationary, the Lipschitz assumption holds with high probability for the compact input domain (e.g., Srinivas et al. (2009)).
    - Assumptions 2 and 4: When the kernel is four times continuously differentiable and stationary, the gradient norm is bounded from above with high probability in a compact input domain (e.g., Srinivas et al. (2009)); however, I do not know of any existing results that suggest the true and approximate Hessian matrices are bounded from above.

Ghosal, Subhashis, and Anindya Roy. "Posterior consistency of Gaussian process prior for nonparametric binary regression." (2006): 2413-2429.
Srinivas, Niranjan, et al. "Gaussian process optimization in the bandit setting: No regret and experimental design." arXiv preprint arXiv:0912.3995 (2009).
Kandasamy, Kirthevasan, et al. "Multi-fidelity Gaussian process bandit optimization." Journal of Artificial Intelligence Research 66 (2019): 151-196.

- There seem to be some gaps between the theoretically and practically verified algorithms without discussion. Discussions about the following gaps are desired:
    - The explanations in Section 3 and the empirical evaluations in Section 6 seem to allow noisy function evaluations; however, the theory and algorithm construction rely on noise-less function evaluation (e.g., Line 8 in Algorithm 1, construction of $m_k$ in Eq.(2)).
    - The treatment of the trust regions is different between practical (Lines 190, 191) and theoretical (Algorithm 2) usages.
    - The random subset-selection of the dimensions (Lines 213-226) seems not to be considered in the theorems.
- The description of the experimental section is inadequate to reproduce the results for the reader. For example, clarification of the following settings is desired:
    - The choice of the kernel.
    - The treatment of the hyperparameters of the kernel.
    - The parameter settings of the local optimization routine.

(Minor)
- Last year's NeurIPS paper (Wu et al. (2023)) is closely related to this paper in the sense that the convergence of the local Bayesian optimization routine is analyzed.

Wu, Kaiwen, et al. "The behavior and convergence of local Bayesian optimization." Advances in Neural Information Processing Systems 36 (2023).

**Questions:**

Is there a possibility to relax Assumptions 1-4? Specifically, I believe that Assumptions 2 and 3 generally do not hold on $\mathbb{R}^D$, even if we rely on probabilistic arguments.

---

> ### Author Response · Authors · 2024-11-28
>
> We sincerely thank for your constructive and trackable feedback. Responses to reviewers' comments are as follows.
>
> Q1: Is there a possibility to relax Assumptions 1-4? Specifically, I believe that Assumptions 2 and 3 generally do not hold on R^D.
>
> A1: We will try to make improvements. Thank you for your advice.

---

### Official Review · Reviewer_Vo6T · 2024-11-01

**Soundness:** 4
**Presentation:** 2
**Contribution:** 3
**Rating:** 5
**Confidence:** 3

**Summary:**

The authors propose an approach to address Bayesian optimisation in high-dimensional spaces. Their method utilises local quadratic models that leverage gradients and Hessians from a global GP and selects new sample points by solving a bound-constrained quadratic program.

**Strengths:**

The paper addresses a relevant problem—high-dimensional Bayesian optimisation—and demonstrates strong performance against various baselines in multiple experiments. Overall, the paper is well written, with a clear discussion of related work and a well-stated contribution. However, the theory section is somewhat disorganised.

**Weaknesses:**

Overall, I find the paper lacks clarity, particularly in the theory section. Each theorem or lemma is presented without any motivation or discussion, making them feel disconnected from the rest of the paper. Additionally, the presentation of the method itself sometimes lacks clarity; for example, the trust region update strategy is barely discussed, including how the trust regions are initialised.

**Questions:**

- How is each trust region's initial radius and point chosen in practice? I think a short discussion about it would help improve the method's clarity.

- I find Section 5 somewhat disorganised. Much of the content could be moved to the appendix, as it does not add significantly to the main paper. In particular, all proofs should be placed in the appendix, which only complicates this section. Moreover, the results are presented without any explanation or intuition. I believe improving this section's readability would improve the paper.

- I think it would be helpful to include a discussion on the chosen trust region update strategy, including whether alternative options were considered and the benefits of the selected. Moreover, what is $\beta_1,\beta_2,\eta_1$ and $\eta_2$ in Algorithm 2? How are they selected?

---

> ### Author Response · Authors · 2024-11-28
>
> We sincerely thank for your constructive and trackable feedback. Responses to reviewers' comments are as follows.
>
> Q1: How is each trust region's initial radius and point chosen in practice? I think a short discussion about it would help improve the method's clarity.
>
> A1: We will supplement relevant discussion in the future. Thank you for your advice.
>
> Q2: I find Section 5 somewhat disorganised. Much of the content could be moved to the appendix, as it does not add significantly to the main paper. In particular, all proofs should be placed in the appendix, which only complicates this section. Moreover, the results are presented without any explanation or intuition. I believe improving this section's readability would improve the paper.
>
> A2: Exactly. We will make revisions. Thank you for your advice.
>
> Q3: I think it would be helpful to include a discussion on the chosen trust region update strategy, including whether alternative options were considered and the benefits of the selected.
>
> A3: We will supplement relevant discussion in the future. Thank you for your advice.

---

### Official Review · Reviewer_NSom · 2024-11-02

**Soundness:** 2
**Presentation:** 1
**Contribution:** 1
**Rating:** 1
**Confidence:** 4

**Summary:**

The paper proposes TuRBO-D, an improvement over previous Trust-Region Bayesian Optimization (TRBO) work. The proposed approach claims to enhance the sampling efficiency of high-dimensional BO tasks over previous TRBO work. The authors present a convergence analysis, showing that their algorithm converges to a stationary point. The authors also conduct experiments comparing their algorithm against previous methods.

The key insight of the algorithm is by leveraging gradients and Hessians of the GP; the underlying objective function can be modeled more accurately, thus leading to sample efficiency compared to TRBO.

**Strengths:**

1. The authors recognize the sample inefficiency of current high-dimensional Bayesian optimization algorithms and propose a more sample-efficient alternative.

2. The authors try to prove the convergence of their proposed algorithm.

3. There is a decent amount of experimental results to demonstrate the validity of the proposed algorithm.

**Weaknesses:**

1. The methods seem to have merit but are hard to validate due to the paper's lack of rigor and presentation clarity.

2. The authors claim their method is more sample-efficient than TRBO. Experimental results show that their method, TRBO-B, is sample efficient and obtains better final values than TRBO (as shown in the bottom row of Fig 1 and Fig 2). Why is that the case? It invalidates all the claims for me. Maybe I am wrong.

3. The paper suffers from issues in presentation, with insufficient explanations of key concepts, unclear notation, and missing justifications for several methodological choices. The cumulative effect of these issues reduces the paper's accessibility and perceived rigor. Here are the main points which make it hard to follow:
  a. The authors claim TRBO suffers from sample inefficiency. Nowhere in the paper is it explained or shown why that is the case.
  b. Why does using gradients and Hessians lead to better sample inefficiency? Why did the authors make this decision?
  c. The algorithm is stated without any explanation. How do the authors choose the final point? It seems like Algorithm 1 returns a point for each trust region.
  d. Baselines and experimental settings are not explained. What is ALEBO? Why did you compare against it? What is Griewank's function, or what is Ackley's?

4. The theoretical results and the actual proofs are stated in the middle of the paper without any supporting text. It is hard to infer what the authors are trying to show and validate its correctness.

Suggestions:
1. Please improve the presentation rigor. Each technical claim and design decision needs to be motivated.
2. The theoretical results need supporting text to explain the math and claims clearly.
3. Experimental settings and results need to be clearly explained.

**Questions:**

1.  How do the authors choose the final point? Algorithm 1 seems to return a point for each trust region. In practice, the output of BO methods is one point that maximizes the acquisition function. It is really hard to infer.

2. How does your method compare against TRBO in terms of runtime? What is cost of calculating gradients and hessian in high-dimensional setting.

3. The authors claim their method is more sample-efficient than TRBO. Experimental results show that their method, TRBO-B, is sample efficient and obtains better final values than TRBO (as shown in the bottom row of Fig 1 and Fig 2). Why is that the case? It invalidates all the claims for me. Maybe I am wrong.

---

> ### Author Response · Authors · 2024-11-28
>
> We sincerely thank for your constructive and trackable feedback. Responses to reviewers' comments are as follows.
>
> Q1: Why does using gradients and Hessians lead to better sample inefficiency? Why did the authors make this decision?
>
> A1: The better sample efficiency benefits from local models sharing the same global GP, instead of gradients and Hessians.
>
> Q2: How do the authors choose the final point?  Algorithm 1 seems to return a point for each trust region. In practice, the output of BO methods is one point that maximizes the acquisition function. It is really hard to infer.
>
> A2: The final point is the minimizer of Eq. 3 and it indeed return a point for each trust region. BO methods usually output a batch of candidates. If batch size = 1, it output one candidate. For our method, the batch size is equal to the number of trust regions.
>
> Q3: How does your method compare against TRBO in terms of runtime?
>
> A3: We will add runtime experiments in the future.
>
> Q4: The authors claim their method is more sample-efficient than TRBO. Experimental results show that their method, TRBO-B, is sample efficient and obtains better final values than TRBO (as shown in the bottom row of Fig 1 and Fig 2). Why is that the case? It invalidates all the claims for me. Maybe I am wrong.
>
> A4: We can't understand your question. We emphasized the reasons for sample efficiency again in answer 1. And the experiments empirically validate the sample efficiency.

---

### Official Review · Reviewer_Td7Q · 2024-11-04

**Soundness:** 2
**Presentation:** 3
**Contribution:** 2
**Rating:** 3
**Confidence:** 4

**Summary:**

This paper proposes TuRBO-D, a trust region-based Bayesian optimization (BO) method for high-dimensional black-box optimization problems. Based on a global Gaussian process (GP) model, TuRBO-D constructs multiple trust regions, and approximate the local function landscape using the quadratic model from GP gradients and Hessians. The points for next round sampling are optimized from the quadratic approximation. The authors provide convergence analysis of the proposed method. The experimental results on both synthetic functions and real-world problems demonstrate that TuRBO-D has better optimization performance than other local BO baselines when the sample budget is less than 300.

**Strengths:**

1. The paper is clear and well-written, and the theoretical part is detailed.

2. The proposed method demonstrate better sample efficiency when the sample budget is small.

**Weaknesses:**

1. I think an important related work [1] is missed by the authors, where the convergence analysis is conducted on a local BO method using GP gradients. The current work seems extend the convergence analysis to the multiple trust region setting.

2. The current experiment results only optimize the problem with a few hundred evaluations, which is far from convergence. I think more evaluations (e.g. a few thousand in original TuRBO paper) can better access the algorithm performance along with the convergence analysis.



[1] Wu, Kaiwen, et al. "The behavior and convergence of local bayesian optimization." Advances in neural information processing systems 36 (2024).

[2] Eriksson, David, et al. "Scalable global optimization via local Bayesian optimization." Advances in neural information processing systems 32 (2019).

**Questions:**

1. The proposed TuRBO-D has two hyperparameters: $\lambda$ for controlling the use of uncertainty in Hessian computation, and $M$ defining the trust region number. I think an ablation study over hyperparameters can help better understanding the contribution of algorithm components.

2. It seems that the trust-region update strategy in Algorithm 2 is different from real implementation mentioned in Line 190. It would be interesting to see the algorithm performance when the implementation is consistent with the theoretical analysis.

3. When the sample budget is small, I think SAASBO [3] is also a strong baseline for high-dimensional optimization. It would help better accessing the algorithm performance when compared against this baseline.

[3] Eriksson, David, and Martin Jankowiak. "High-dimensional Bayesian optimization with sparse axis-aligned subspaces." Uncertainty in Artificial Intelligence. PMLR, 2021.

---

> ### Author Response · Authors · 2024-11-28
>
> We sincerely thank for your constructive and trackable feedback. Responses to reviewers' comments are as follows.
>
> Q1: The proposed TuRBO-D has two hyperparameters \lambda for controlling the use of uncertainty in Hessian computation, and M defining the trust region number. I think an ablation study over hyperparameters can help better understanding the contribution of algorithm components.
>
> A1: Exactly. We will conduct relevant experiments. Thank you for your advice.
>
> Q2: It seems that the trust-region update strategy in Algorithm 2 is different from real implementation mentioned in Line 190. It would be interesting to see the algorithm performance when the implementation is consistent with the theoretical analysis.
>
> A2:  Exactly. If possible, we will try to improve this.
>
> Q3: When the sample budget is small, I think SAASBO [3] is also a strong baseline for high-dimensional optimization. It would help better accessing the algorithm performance when compared against this baseline.
>
> A3: Exactly. We will add relevant  experiments.

---

### Official Review · Reviewer_gLXE · 2024-11-04

**Soundness:** 3
**Presentation:** 3
**Contribution:** 3
**Rating:** 5
**Confidence:** 3

**Summary:**

The paper introduces an advanced approach for high-dimensional Bayesian Optimization (BO) using derivatives to improve sampling efficiency in trust-region-based BO frameworks. The method constructs local quadratic models using gradients and Hessians derived from a global Gaussian Process (GP), facilitating efficient optimization across trust regions and achieving better sampling efficiency without relying solely on local GPs.

**Strengths:**

1. The paper offers convergence proofs enhancing the method’s robustness.
2. The proposed method addresses the dimensionality issue in BO.
3 The experimental evaluation shows that TuRBO-D outperforms existing BO methods on tasks with up to 300 dimensions.

**Weaknesses:**

1. The complexity experiments are not included in the paper
2. The code has not been found in the paper

**Questions:**

* Since the method includes the second-order gradients, how would this impact the general complexity of the algorithm?
* Will a higher-order derivative help more? If so, how do we specify the order we should use?
* How would including a second-order gradient differentiate first-order gradient-only methods? Can we conduct an ablation regarding this?
* Will the provided method be sensitive to predefined parameters?

---

> ### Author Response · Authors · 2024-11-28
>
> We sincerely thank for your constructive and trackable feedback. Responses to reviewers' comments are as follows.
>
> Q1: Since the method includes the second-order gradients, how would this impact the general complexity of the algorithm?
>
> A1: On the one hand, in practice, we use pytorch to automatically calculate derivatives, and the overall running time of our algorithm is lower than that of TuRBO. On the other hand, the complexity depends on the dimension of the problem. The dimension of experiments is around 100, which does not lead to expensive overhead to find derivatives. As for higher dimensions, we are still untested.
>
> Q2: Will a higher-order derivative help more? If so, how do we specify the order we should use?
>
> A2: I think higher derivatives are not necessary. The second derivative already gives the curvature information. Besides, in the field of numerical optimization, one usually uses the first and second derivatives, such as L-BFGS, CG, and so on. Higher derivatives also cause the model to overfit.
>
> Q3: How would including a second-order gradient differentiate first-order gradient-only methods? Can we conduct an ablation regarding this?
>
> A3: We will add ablation experiments later.
>
> Q4: Will the provided method be sensitive to predefined parameters?
>
> A4: We didn't explore sensitivity. If possible, we will conduct relevant experiments. Thank you for your advice.

---

### Official Review · Reviewer_Tt1x · 2024-11-09

**Soundness:** 2
**Presentation:** 3
**Contribution:** 2
**Rating:** 5
**Confidence:** 3

**Summary:**

This paper proposes an approach for high dimensional Bayesian optimization, which the authors call TuRBO-D, that takes the posterior mean and variance of the fitted Gaussian process (GP) to form local quadratic functions as a proxy for acquisition. Instead of doing local GP  fit as in TuRBO which suffers from a reduction of sampling efficiency, TuRBO-D utilizes statistical information from a global GP fit to invoke a trust region mechanism to handle high dimensionality. The authors provide a theoretical convergence result and conduct a range of experiments to support their approach.

**Strengths:**

- While there have been work in injecting derivative information from the (expensive) objective function into posterior GP fitting, the approach carried out by the authors, namely extracting posterior GP to inform the derivatives in constructing local quadratic approximation, appears novel and has computational benefits.

- The improvements over benchmarks like TuRBO are demonstrated in the range of experiments.

**Weaknesses:**

1. From my understanding and as the authors also describe in their paper, a main strength and motivation of TuRBO is to allow for the heterogeneous modeling of the objective function which is especially important in high dimension. This is done by local GP fits. On the other hand, the authors' approach globally fits the GP which seems to basically forgo this important strength of TuRBO. In other words, if the global GP fit is bad, then the extracted derivatives are also badly informed, and thus the method would work poorly in high dimension. The authors claim that their approach gains sampling efficiency, but how important is this compared to the heterogeneity in modeling the objective? It seems that heterogeneity is more important, as the GP fit typically depends negligibly on points that are far away but a poor modeling of the objective function is more impactful (please correct me if I'm wrong).

3. The main theoretical result (Theorem 4) is weak in the following sense: It says, with high probability, the inf of gradient over the solution sequence is close to 0. However, this inf only means that there exists a subsequence whose gradients are close to 0. If we stop the algorithm at some large iteration step, there is no guarantee that the gradient is small since that step may not be in that subsequence. That is, even putting aside the lack of convergence rate etc. (which I understand can be difficult to obtain in general for BO algorithms) and viewing at a basic level, the theoretical result does not seem meaningful.

The above are my main concerns. Additionally,

1. The authors propose to use a subset of dimensions instead of full dimension to form the quadratic approximation, but the theoretical analyses do not take this into account.

2. A possible strength of TuRBO is that the local fit uses less computation due to the smaller matrix inversion. However, by global fitting, the authors' approach faces a computation load as high as standard BO (as least for this posterior update step). This might need some discussions.

3. The implementation of the main benchmark TuRBO has no details, and it's unclear if the comparison is fair. E.g., how do you set the hyperparameters for TuRBO and TuRBO-D?

**Questions:**

Line 179: It'd be helpful to discuss how to choose the hyperparameter lambda or at least provide some intuition.

Line 214: It'd be helpful to discuss whether vanishing derivative is a well-known issue of GP in high dimension. If it's your discovery, then please clarify. Moreover, it would be good to clarify whether the approach of using a subset of dimensions is known in addressing this issue or is your invention.

Line 400: The epsilon is not used in the conclusion of the theorem.

---

> ### Author Response · Authors · 2024-11-28
>
> We sincerely thank for your constructive and trackable feedback. Responses to reviewers' comments are as follows.
>
> Q1: It'd be helpful to discuss how to choose the hyperparameter lambda or at least provide some intuition.
>
> A1: We're still not sure how to tune the parameter lambda better. Here, we just treat lambda as a small noise epsilon.
>
> Q2: It'd be helpful to discuss whether vanishing derivative is a well-known issue of GP in high dimension.
>
> A2: We are not clear whether it is a well-known issue. But this is indeed the problem we encounter in practice.
>
> Q3: The epsilon is not used in the conclusion of the theorem.
>
> A3: Exactly. We will make corrections later.

---

### Official Review · Reviewer_Umyf · 2024-11-09

**Soundness:** 3
**Presentation:** 3
**Contribution:** 3
**Rating:** 6
**Confidence:** 3

**Summary:**

This paper proposes the TuRBO-D method for optimizing expensive blackbox functions, which improve on the previous work of TuRBO (trust region Bayesian optimization) by using local quadratic models within each trust region and enable applications to mid/high-dimensional settings. Theoretical results are provided to justify the convergence of this method to a stationary point of the true underlying blackbox functions. Simulations and some real world applications are provided.

**Strengths:**

1. The paper is overall clear and well-written.
2. The proposed method, originating from TuRBO, has a good and clear motivation.
3. Necessary background materials are provided.
4. Theoretical analysis of convergence to a stationary point are helpful.
5. The performance of the proposed method are justified through a few real world applications.

**Weaknesses:**

To me, it seems that more background and discussions on the original TuRBO method (Eriksson et al., 2019) could be provided. For instance, TuRBO method and the newly proposed TuRBO-D method has the following differences, which would be helpful to discuss the motivations and reasons:
1. TuRBO uses local GP within each trust region to deal with large number of observations, whereas TuRBO-D does local quadratic approximation to a glocal GP instead of local GPs
2. TuRBO choose samples across trust regions through Thompson sampling, whereas TuRBO-D traverse through all trust regions recursively, each time choosing a sample from a trust region.

There are many real world applications in the TuRBO paper (Eriksson et al., 2019). While the current applications do provide certain convincing justifications, it would be helpful to also see how TuRBO-D performs in applications of the TuRBO paper.

**Questions:**

What would be some either rigorous or intuitive justifications for choosing g_k = \nabla \mu_k(x_k）and B_k = \nabla^2 \mu_k(x_k) + \lambda \nabla^2 \sigma_k(x_k) in line 177? How to tune the hyperparameter \lambda in applications? And how does the choice of \lambda affect the convergence analysis in Sec 5?

---

> ### Author Response · Authors · 2024-11-28
>
> We sincerely thank for your constructive and trackable feedback. Responses to reviewers' comments are as follows.
>
> Q1: How to tune the hyperparameter \lambda in applications?
>
> A1: We're still not sure how to tune the parameter lambda better. Here, we just treat lambda as a small noise epsilon.
>
> Q2: How does the choice of \lambda affect the convergence analysis in Sec 5?
>
> A2: The choice of the \lambda does not affect the convergence analysis, because the analysis depends only on the boundedness of Bk.

---

### Note · Authors · 2024-11-28

I have read and agree with the venue's withdrawal policy on behalf of myself and my co-authors.